# The Study of the Impact of Carbon Finance Effect on Carbon Emissions in Beijing-Tianjin-Hebei Region—Based on Logarithmic Mean Divisia Index Decomposition Analysis

**Li Li [1], Di Liu [1], Jian Hou [2],\*, Dandan Xu [1] and Wenbo Chao [1]**

[1]  School of Economics, Beijing Technology and Business University, Beijing 100048, China; lililly@th.btbu.edu.cn (L.L.); 13126630135@163.com (D.L.); lxxddnn@163.com (D.X.); tongjicwb@163.com (W.C.)

[2]  School of Economics and Management, Beijing Forestry University, Beijing 100083, China

\*  Correspondence: houjian1128@bjfu.edu.cn

**Abstract:**  The negative effects of global warming are becoming more and more serious. The fundamental way to prevent global warming is by reducing carbon dioxide emissions. Achieving this has become a key concern for all countries. The logarithmic mean divisia index model was constructed to decompose the total carbon emission increment. Carbon finance effect was divided into green credit effect and carbon trading effect to analyze the impact of carbon finance on carbon emissions. The results showed that the total carbon emission reduction value caused by green credit effect from 2010 to 2016 in the Beijing-Tianjin-Hebei region was 66193.96 million tons, and the added value of carbon emission caused by carbon trading effect was 80266.68 million tons. There are regional differences in the effects of carbon finance on carbon emissions in these regions. It can be concluded that to a certain extent, green credit can reduce carbon emissions, and carbon trading can increase carbon emissions. Using the gradual expansion of carbon finance trading and market mechanism of carbon finance to solve the problem of carbon emission can improve the efficiency of carbon emission reduction.

**Keywords:** carbon finance effect; carbon emissions; logarithmic mean divisia index (LMDI) model; green credit; carbon trading

## 1. Introduction

The abnormal climate change over the past 100 years has caused severe challenges and threats to the survival and development of human society. According to the Intergovernmental Panel on Climate Change, the amount of carbon dioxide in the atmosphere has increased by 2.0 parts per million (PPM) per year [1]. In 2014, global carbon emissions reached 36.14 billion tons, which is about three times more than the amount of global carbon emissions in 1960 [2]. It is widely recognized that carbon emissions have caused global warming, which is the most serious challenge to human survival and development [3]. The global warming caused by greenhouse gases has attracted wide attention from people worldwide. The fundamental way to prevent global warming is to reduce carbon dioxide emissions [4]. The United Nations Framework Convention on Climate Framework shows that the world has begun to control carbon dioxide emissions in action [5].

On the one hand, China's economy is growing rapidly and has become the world's second-largest economy in the world [6,7]. On the other hand, the problem of carbon emissions in China is serious. China's economic growth in recent decades is mainly based on an extensive growth model at the

expense of high input, high consumption and high pollution, while the benefit generated by this model is low. Since 2008, the amount of China's carbon dioxide emissions has ranked first in the world, with the per capita carbon dioxide emissions up to 7.2 tons in 2013, while the EU per capita emissions of 6.8 tons in 2008 [8]. In 2016, China emitted 28% of the world's total carbon emission [9]. Some have argued that the goal of controlling global warming will be very difficult to achieve unless the speed of emissions slow down in China [10].

With the acceleration of industrialization and urbanization, dependence on energy has been increasing. It can be seen that the "three highs-one low" extensive growth model needs to be transformed into a low-carbon sustainable growth model. Carbon emissions has become a practical constraint to China's economic growth, as the 12th Five-Year national plan incorporated carbon emission intensity into the assessment system as a binding indicator of social development. The 13th Five-Year Plan Draft proposed to take initiatives to control carbon emissions, fulfill emission reduction commitments, effectively control greenhouse gas emissions, and comprehensively promote green development. In 2015, the Notice of the State Council on Printing and Distributing the National Standardization System Construction and Development Plan (2016–2020), a document issued by the State Council, clearly stated that standardization of ecological civilization should be strengthened, and the development of carbon emission, energy conservation and environmental protection industries and circular economy guidelines should be accelerated to effectively improve the level of low-carbon economy. At the same time, China pointed out that the government has integrated climate change into the overall economic and social development strategy of the country. Carbon finance is the best way to relieve the pressure of the task of carbon emission reduction [4]. The European Union, the most developed region in the world, in terms of carbon finance transactions, has indicated that carbon finance transactions can fully exploit the price discovery function of greenhouse gas emissions and use market mechanisms to curb carbon emissions. Therefore, carrying out carbon finance transactions can help China achieve the strategic goal of a green economy, and ultimately fulfill the commitment of emissions reduction commitments of China. It is imperative to establish a carbon financial system. However, how to carry out carbon finance transactions in China and how to carry out carbon finance transactions in regions with large differences in regional development level is an urgent problem needed to be solved in China, which is also the value and significance of this paper.

It is exactly because of this background, coupled with the continuous improvement of China's economic level and people's life quality, that people's requirements for the environment are increasingly getting higher. Therefore, it is of theoretical and practical significance to analyze the impact of carbon finance on carbon emission reduction in various regions.

In this study, taking the Beijing-Tianjin-Hebei region as an example, the logarithmic mean divisia index (LMDI) model is constructed to analyze the influencing factors of carbon emissions in Beijing, Tianjin and Hebei, and to analyze the regional differences of the effects of carbon finance in these three regions, so as to provide some ideas, methods and policy suggestions for the construction of China's carbon financial system.

## 2. A Brief Overview of Literature

### 2.1. The Cost of Carbon Reduction

In the existing literature, carbon reduction cost has different connotations and expressions [11], including marginal emission reduction cost, average emission reduction cost, shadow price and carbon price. Ellerman and Decaux [12] used the emissions prediction and policy analysis (EPPA) model to obtain the marginal emission reduction cost curve of 12 countries and regions in the world, and found that the relationship between marginal emission reduction cost and emission reduction is a quadratic function. Schelling [13] suggested that the cost of emission reduction in developing countries is far higher than that of developed countries, and the latter should provide adequate capital and technologies to support developing countries and shoulder more obligations. Goulder

et al. [14] showed that the non-auction quota approach leads to the highest costs through general equilibrium analysis. Chaurey and Kandpal [15] put forward that if the Indian government used the "solar home system" supplemented by carbon finance, it can significantly reduce costs. Wu et al. [16] simulated the marginal emission reduction cost curve of various provinces and cities by building a multi-region dynamic general equilibrium model in China, and found that there were differences between the upturning range and inflection point position of the marginal emission reduction cost curve of different provinces and cities. Ma et al. [17] pointed out that the impact of financial deepening on carbon emissions results in obvious regional differences. Rahman and Kirman [18] simulated the relationship between the emission reduction costs and emission reduction of 13 clean development mechanism (CDM) projects in four countries, including China, and found that the emission reduction costs of different countries and different types of projects were significantly different.

### 2.2. The Impact of Financial Development on Carbon Emissions

Some studies are conducted from the perspective of financial development, playing its role in capital allocation to support carbon emission reduction. Linares and Perez-Arriaga [19] argued that the key for developing and underdeveloped countries to use low-carbon technologies to cope with global warming was the need for financial support from developed countries. Shahbaz [20] found that financial development provided financing channels for emission reduction technologies, so as to guide the optimization and upgradation of energy structure and promote carbon emission reduction.

Different conclusions have been drawn on the impact of financial development on carbon emissions. Sadorsky [21] used the GMM model to test the impact of financial development in Pakistan on carbon emissions and found a negative correlation between them. On one hand, financial development enables enterprises to reduce financing costs, increase production scale, and increase carbon emissions; on the other hand, it enables consumers to increase their consumption of automobiles due to easier access to credit. Jalil and Feridum [22] argued that China's financial development had a significant negative correlation with carbon emissions. Gu and He [23] found that regional financial deepening on carbon emissions exerts a significant inhibitory effect. Yan et al. [24] found a u-shaped correlation between the impact of financial development on carbon dioxide emissions. Xiong and Qi [25] showed that financial development had a stimulative effect on carbon emissions, and financial innovation and development should be constantly strengthened to improve the degree of integration between financial development policies and carbon emissions. Shao and Liu [26] found that financial development restrained carbon emissions to a certain extent, but the effect of emission reduction would change with the alteration of other carbon emission factors. Therefore, the advantages of financial development in carbon emission reduction should be highlighted to promote the development of a low-carbon economy.

### 2.3. The Construction of Carbon Finance Market

Yang and Chen [27] suggested that regional carbon trading market should be built up, but the regional development levels are quite different and it is unrealistic to set up a unified "carbon trading" market in the short term. Chen and Liu [28] pointed out that diversity will be an important feature of the construction of the carbon market. The eastern region, relatively developed, can refer to developed countries with an emission reduction policy. The western region should take a more moderate policy to encourage the economy to gradually transform into low-carbon economy without affecting economic development. Zhang [29] proposed that the three elements for China to establish a carbon trading market are products, mechanism, and participants. Yang and Zhang [30] suggested that the carbon financial system should be built by improving the system and infrastructure and strengthening the innovation of carbon financial instruments. Mei and Xu [31] came up with China's carbon financial system framework in three aspects, including carbon financial institution, carbon financial market and carbon financial system, among which carbon financial market consists of a trading platform, trading mechanism and trading products. Zhang [32] proposed to promote the construction of China's

carbon finance development system from four aspects: improving carbon trading financing methods, establishing risk management system, developing carbon futures contracts and participating in carbon trading intermediary services.

### 2.4. The Stock Market in Promoting Renewable or Clean Energy Projects and Reducing Carbon Emissions

Paramati et al. [33] confirm that stock market has a negative and positive effect on the $CO_2$ emissions of developed and developing economies, respectively. The non-renewable energy contributes to higher $CO_2$ emissions, while renewable energy reduces $CO_2$ across the developed and developing economies of the Group of Twenty (G20). Paramati et al. [34] further finds that the development of the stock market and economic growth increase energy consumption, and the four driving forces of energy consumption are gross domestic product, stock market development, industrialization and internationalization. The findings indicate that both stock market and economic development increase energy consumption. Their findings reflect that African frontier market economies need to balance foreign investment, trade and stock market activities with better energy integration for sustainable development. In this respect, the role of stock market will play a major role in attracting institutional investors to invest in frontier markets. Kutan et al. [35] argued that among major emerging market economies, not only stock market indicators, but also Foreign Direct Investment (FDI) inflows play an important role in promoting renewable energy consumption. Paramati et al. [36] further investigate the extent to which stock market developments affect the clean energy consumption of the G20, European Union (EU) and Organization for Economic Co-operation and Development (OECD) countries. The results showed that stock market developments played a significant role in promoting clean energy across all national country groups. Based on relevant studies, Paramati et al. [37] found that stock market indicators have a different relationship with $CO_2$ emissions in developed and emerging market economies; the growth of stock markets in developed countries is substantially reducing $CO_2$ emissions, while it is increasing them in emerging economies.

On the whole, more research is being conducted at the macro level of the country, making quantitative analysis by establishing mathematical models. At present, there are few specific studies on the development of carbon finance in various regions in China, but there are great differences in development between regions. It is very difficult to find a development model that is universally applicable across the country. Therefore, studying the effects of carbon finance in different regions to support carbon emission reduction is important to promote China's carbon financial trading system.

## 3. Methodology and Data

### 3.1. Constructing an LMDI Model

Ang [38] first proposed the logarithmic mean divisia index (LMDI), which is the addition form of LMDI, a completely non-residual decomposition method. Ang [39] proved that the multiplicative form of this method can make up for the shortcomings of previous research methods. At present, the method is mainly used in the field of energy. The basic idea of this approach is to decompose the target values into several main influencing factors based on mathematical identity transformation, and the method mainly decomposes from the different layers to the total increment, thereby quantifying the contribution of the changes of the structure at different levels to total incremental change. This study attempts to separate carbon trading and green credit changes from various factors and analyze their impacts on carbon emissions.

Using the LMDI method to conduct research is a decision based on the consideration of two aspects: On one hand, this method can accurately quantify the contribution of carbon emission increments from structural changes at different levels. When the impact on carbon emissions is decomposed into carbon trading and green credits, traditional linear regression methods are unable to reduce the interference of residuals to carbon trading and green credits to acceptable levels. Therefore, the impact of carbon financial effects on carbon emissions cannot be fully and accurately explored.

The use of the LMDI method can fully explain the impact of the two on carbon emission increments, that is, all the carbon emission increments are allocated to the two factors of carbon trading and green credit. On the other hand, the LMDI method uses the decomposition method instead of the classical regression analysis method, which can effectively avoid the "pseudo-regression" problem caused by the traditional linear regression analysis method.

In this study, the LMDI method is adopted to decompose carbon emissions according to the carbon finance technique, which is divided into green credit and carbon trading to build the total carbon emission decomposition model. It is assumed that the factors that may affect carbon emissions are gross output value ($Y$), carbon emission reduction ($CER$), and total green credit ($Cre$). The formula is as follows:

$$C = \sum_i \frac{C_i}{Cre_i} \times \frac{Cre_i}{Y_i} \times \frac{Y_i}{CER_i} \times CER_i = \sum_i U_i \times Q_i \times D_i \times CER_i \tag{1}$$

where $i$ represents the region, $i = 1$ for the Beijing; $i = 2$ for the Tianjin; $i = 3$ for the Hebei; $U = \frac{C}{Cre}$ indicates the carbon emissions supported by unit green credit, and measures the degree of economic low carbonization of credit; $Q = \frac{Cre}{Y}$ represents the total amount of green credit needed by the unit output value, measuring the intensity of green credit; $D = \frac{Y}{CER}$ stands for the contribution of carbon emission reductions of the CDM project, and measures the capacity of CDM projects to reduce carbon emissions. Carbon finance support is made up of green credit and carbon trading support.

The LMDI is divided into multiplicative decomposition and additive decomposition, both of which are equally valid and can be transformed into each other. In contrast to the multiplicative case, the decomposition results of the additive case are given as physical units rather than indexes, making them easy to use and interpret. In this way, additive decomposition is used to decompose the carbon emissions. The effects of the four factors that contribute to changes in carbon emissions over $t$ years are represented by $\Delta C_u$, $\Delta C_q$, $\Delta C_d$ and $\Delta C_{cer}$. The total change in carbon emissions from the year $t-1$ to year $t$ is as follows:

According to the addition of LMDI decomposition, from $t-1$ to $t$ time, the total increase in carbon dioxide emissions can be expressed as:

$$\begin{aligned} \Delta C = C^t - C^{t-1} \quad &= \sum_i U_i^t \times Q_i^t \times D_i^t \times CER_i^t - \sum_i U_i^{t-1} \times Q_i^{t-1} \times D_i^{t-1} \times CER_i^{t-1} \\ &= \Delta C_u + \Delta C_q + \Delta C_d + \Delta C_{cer} + \Delta C_{resid} \end{aligned} \tag{2}$$

where $\Delta C$ is the total change in carbon emissions, $C^t$ and $C^{t-1}$ are carbon emissions in years $t$ and $t-1$, respectively, in the study area, $\Delta C_u$ is the change in carbon emissions caused by changes in the green credit support, $\Delta C_q$ is the change in carbon emissions caused by green credit expansion, $\Delta C_d$ is the change in carbon emissions caused by CDM project, $\Delta C_{cer}$ is the change in carbon emissions caused by carbon trading, and $\Delta C_{resid}$ represents the decomposition margin.

Among them, $\Delta C_1 = \Delta C_u + \Delta C_q$, $\Delta C_2 = \Delta C_d + \Delta C_{cer}$.

$\Delta C_1$ represents the total green credit effect, $\Delta C_2$ represents the total carbon trading effect.

Based on the additive decomposition model, the general formulae for carbon emissions decomposition can be summarized as follows:

$$\Delta C_u = \sum_i W_i^t In \frac{U_i^t}{U_i^{t-1}} \tag{3}$$

$$\Delta C_q = \sum_i W_i^t In \frac{Q_i^t}{Q_i^{t-1}} \tag{4}$$

$$\Delta C_d = \sum_i W_i^t In \frac{D_i^t}{D_i^{t-1}} \tag{5}$$

$$\Delta C_{cer} = \sum_i W_i^t In \frac{CER_i^t}{CER_i^{t-1}} \tag{6}$$

$$W_i^t = \frac{C_i^t - C_i^{t-1}}{InC_i^t - InC_i^{t-1}} \tag{7}$$

where $W_i^t$ represents logarithmic weight. As shown in Equations (3)–(6), a positive value indicates that the factors increase carbon emissions. By contrast, a negative value indicates that the factors decrease carbon emissions.

### 3.2. Basic Data

The data for carbon emissions are derived from the carbon emission reference coefficient multiplied by the total final energy consumption. The reference coefficient is taken from the China Contract Energy Management Network. The final energy consumption is divided into 10 categories, which are coke, coal, crude oil, diesel, kerosene, gasoline, fuel oil, natural gas, refinery dry gas and liquefied petroleum gas.

The data of energy consumption, total green credit (*Cre*), gross output value (*Y*), and carbon emission reduction (*CER*) of the three regions are respectively from the regional energy balance sheet in China Energy Statistical Yearbook, China Financial Yearbook and Annual Social Responsibility Report with China's five largest banks (Industrial and Commercial Bank of China, Agricultural Bank of China, Bank of China, China Construction Bank and Bank of Communications), China Statistical Yearbook, and estimated annual emission reductions for projects approved by provinces and municipalities in the CDM project database system in the China Clean Development Mechanism Network. The green credit amount of these three regions is estimated based on the green credit amount of the five major banks in the country multiplied by the proportion of credit in each region to the national credit.

## 4. An Empirical Analysis of the Effects of Carbon Finance on Carbon Emissions

### 4.1. An Empirical Study on the Impact of Carbon Finance Effect on Carbon Emissions from the Beijing-Tianjin-Hebei Region

Using the LMDI addition model to structurally decompose the impact of carbon finance effects on carbon emissions in the Beijing-Tianjin-Hebei region, the results of carbon emission increments from green credit (*CRE*) and carbon emission reduction (*CER*) decomposition are shown in Table 1:

**Table 1.** Calculation result of LMDI addition in three regions (Unit: ten thousand tons).

| Year | $\Delta C$ | $\Delta C_u$ | $\Delta C_q$ | $\Delta C_1$ | $\Delta C_d$ | $\Delta C_{cer}$ | $\Delta C_2$ |
|------|------|------|------|------|------|------|------|
| 2010 | 9438.72 | −11,214.65 | 1906.02 | −9308.63 | 51,145.45 | −32,398.10 | 18,747.35 |
| 2011 | 11,562.49 | −10,849.28 | 559.77 | −10,289.51 | −69,941.03 | 91,793.02 | 21,851.99 |
| 2012 | 1645.84 | −6839.19 | −2966.30 | −9805.49 | −41,509.76 | 52,961.08 | 11,451.33 |
| 2013 | −673.35 | −38,564.35 | 27,866.81 | −10,697.54 | 313,758.78 | −303,734.58 | 10,024.20 |
| 2014 | −4915.56 | −17,580.54 | 6882.57 | −10,697.97 | 89,836.97 | −84,054.56 | 5782.41 |
| 2015 | −1390.25 | −35,588.53 | 31,078.91 | −4509.61 | −67,851.01 | 70,970.37 | 3119.36 |
| 2016 | −1595.16 | −15,823.34 | 4938.13 | −10,885.21 | −31,852.09 | 41,142.14 | 9290.05 |
| 2010-2016 | 14,072.73 | −136,459.87 | 70,265.92 | −66,193.96 | 243,587.31 | −163,320.62 | 80,266.68 |

### 4.1.1. The Effects of Green Credit on Carbon Emissions

From 2010 to 2016, green credit was the main inhibitor of carbon emissions in the three regions. The total carbon emission reduction caused by green credit is 661,939,600 tons, and it has a negative effect on carbon emissions every year, indicating that green credit can reduce carbon emissions to a certain extent. The total contribution of green credit support for carbon emissions was −136,459.87 million tons, indicating that green credit has a negative effect on carbon emissions as a whole. And from the sub-terms, the contribution value of each year was negative, suggesting that

it had a negative effect on carbon emissions each year. However, in contrast, the carbon emissions contributed by the green credit expansion effect are 70,265.92 million tons; in 2015, the contribution value reached 310,789,100 tons, accounting for 44.23% of the total, which shows that the green credit expansion effect on the carbon emissions has a positive effect on the whole (most obvious in 2009). However, in terms of annual values, the carbon emissions caused by this effect are either positive or negative, and relatively equal, which shows that the adjustment of this factor by the carbon emission reduction effect is not obvious.

### 4.1.2. The Effects of Carbon Trading on Carbon Emissions

From 2010 to 2016, carbon trading was a major contributor to carbon emissions in the Beijing-Tianjin-Hebei region. The carbon emissions contribution from carbon trading is 80,266.68 million tons, and it has a positive effect on carbon emissions every year, indicating that carbon trading can increase carbon emissions to a certain extent. The carbon emission reduction effect contributions and carbon trading effect contributions of CDM projects were 243,587,310 tons and 1,633,206,200 tons, respectively. The results show that carbon emission reduction effect of CDM projects has a positive effect on carbon emission, while carbon trade effect has a negative effect on carbon emission. From the numerical values of each year, the carbon emission contribution effects and carbon trading effects of CDM projects have positive and negative values, but the former is more positive and the latter has more negative values, which shows that the carbon emission reduction effect caused by the adjustment of these two is not obvious, and the former is more likely to increase carbon emissions, while the latter is more likely to reduce carbon emissions.

### 4.2. An Empirical Analysis of the Effects of Carbon Finance on Carbon Emissions in the Sub-Region

In order to understand the effects of carbon finance on carbon emissions in depth, it is necessary to analyze the contribution of the total carbon trading effect and the total green credit effect to carbon emissions of each region, and then obtain the difference between the three regions. The specific analysis is as follows:

### 4.2.1. Contribution of the Total Green Credit Effect of Each Region to Carbon Emissions

The contribution of the total green credit effect to carbon emissions in each region from 2010 to 2016 is shown in Figure 1. It can be seen from Figure 1 that the total green credit contribution of the three major regions has a negative contribution to carbon emissions, indicating that the total green credit effect can suppress the increase of carbon emissions. From a dynamic point of view, the changes in carbon contribution values generated by the total effect of green credits in the three regions are not synchronized. The three regions are only synchronized in 2016, and the remaining years are not synchronized. Taking 2011 as an example, carbon emissions in Hebei increased, while the other two regions declined.

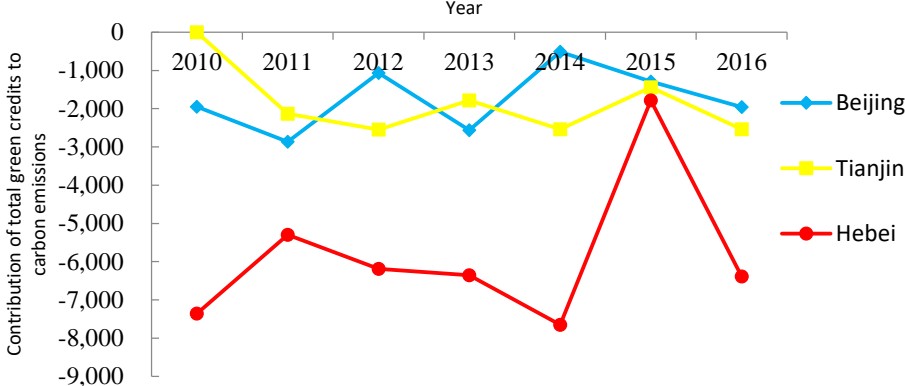

**Figure 1.** Contribution of total green credits to carbon emissions in each region from 2010 to 2016.

From the perspective of regional structure, green credit plays the most significant role in promoting carbon emission reduction in Hebei and can best reflect the effect of carbon emission reduction, as this may be related to the high proportion of Hebei's secondary industry. From the perspective of the three industries, carbon emissions are mainly generated by the secondary industry, and green credit support has greatly inhibited the carbon emissions of the secondary industry in Hebei region. It can be seen intuitively from the figure that the vertical axis of the Hebei area is at the bottom. Beijing and Tianjin are intertwined, and the difference between the two is small, and above Hebei, indicating that the carbon emission reduction effect is weaker than that of Hebei.

The Beijing-Tianjin-Hebei region should formulate industrial upgrading plans based on the industrial development of the region and rationally allocate the proportion of the three industrial structures. The situation of high pollution and high emissions in the Beijing-Tianjin-Hebei region is relatively serious, especially in Hebei Province, which is closely related to the high proportion of high-pollution and high-emission industries in the region, the high proportion of the secondary industry, and the slow development of the tertiary industry. To achieve low-carbon development, industrial structure transformation and upgradation must be carried out. On one hand, reduce the proportion of the secondary industry, promote industrial transformation; and on the other hand, increase the application of high and new technology in industry, reduce industrial pollution and emissions, and accelerate the pollution control and emission reduction governance by introducing advanced technology to reduce Beijing-Tianjin-Hebei Regional carbon emissions.

The empirical analysis above shows that green credit can help reduce $CO_2$ emissions. At present, China's green credit is still at an initial stage of development. By 2014, the banking sector's green credit line had reached 7.59 trillion yuan, supporting a total of 15,718 energy conservation and environmental protection projects, and achieved good economic returns and results. By the end of 2015, the balance of green credit project loans of major banking financial institutions was more than 7 trillion yuan, an increase of 16.47% over the same period. However, China's green credit development faces many challenges, such as the imperfect green credit risk evasion mechanism and the imperfect environmental information communication mechanism.

### 4.2.2. Contribution of the Total Carbon Trading Effect of Each Region to Carbon Emissions

The contribution of the total carbon trading effect to carbon emissions in each region from 2010 to 2016 is shown in Figure 2. It can be seen from Figure 2 that the carbon emission contribution value of total carbon trading effect of the three regions is positive, which means that the total effect of carbon trading can promote the increase of carbon emission. That is, carbon trading promotes the development of CDM projects. The more active the carbon trade, the more the carbon emissions are. There are two reasons: First, carbon emissions have a transfer effect—developed countries cooperate with developing countries by providing funds and technology, "Funding + Technology" in exchange for greenhouse gas emission rights. Due to the different emissions and emission reduction costs of different enterprises, some enterprises with more emission rights can sell redundant indicators to enterprises with insufficient emission rights. Second, there is a lack of self-discipline in carbon emissions. Green development and low-carbon development are the future directions of the trend of the times. However, a considerable number of enterprises are seriously short of investment in environmental protection in order to maximize their own interests. The phenomenon of waste gas stealthily discharging occurs from time to time, and the self-discipline effect is lacking. As a result, many enterprises are not included in the carbon emission trading system. The above two factors have led to an increase in carbon trading, but carbon emissions have increased.

Carbon finance has just started in China. The country adopts a voluntary emission reduction method. The property rights of carbon emission rights are not clear, weakening the property rights of carbon emission rights, and there is no mandatory binding force, which reduces the purchase intention of buyers.

From a dynamic point of view, the general trend of the three regions is: rising before 2011, slipping from 2011 to 2015 and rising from 2015 to 2016. During the period, Hebei is up sharply. From the perspective of regional structure, the carbon trading effect caused by Hebei's carbon trading effect is the most obvious, and Tianjin is slightly higher than Beijing, the gap between the two places is not large. It can also be seen from the figure that the vertical axis of the Hebei area is always at the top, while the vertical axis of Beijing is at the bottom, and Tianjin is in the middle.

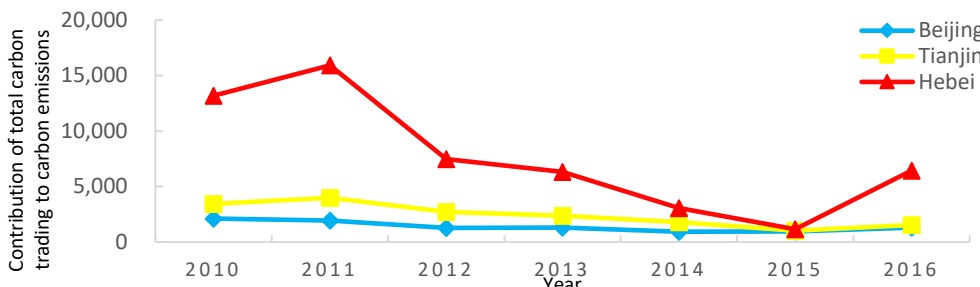

**Figure 2.** Contribution value of total carbon trading effect to carbon emissions in each region from 2010 to 2016.

At the end of 2014, Beijing and Hebei took the lead in launching a cross-regional carbon emissions trading pilot. Cross-regional carbon trading can not only expand the capacity of Beijing's carbon trading market, but also increase market activity. It can achieve cross-industry and cross-regional ecological compensation through marketization mechanism. The implementation of the carbon trading in the Beijing-Tianjin-Hebei region is slow, mainly due to the progress of all links from management to quota allocation to late punishment, and the standard Beijing-Tianjin-Hebei is difficult to unify. From the pilot to the regional synergy, the system construction plays a pivotal role. Therefore, on the existing basis, the carbon quota allocation and auction system needs to be further refined, and the benign development of the carbon market should be promoted through institutional construction.

It can be concluded from the current situation of the Beijing-Tianjin-Hebei trans-regional carbon trading market system that carbon trading can only fully exert its emission reduction effect by breaking restrictions and realizing cross-regional transactions. According to the characteristics of different regions, a multi-combined carbon trading market system will be constructed, and on this basis, the formation of a unified national carbon trading platform will be promoted in an orderly manner. The national unified carbon trading platform should be gradually improved. The seven carbon trading pilots in Beijing, Shanghai, Tianjin, Shenzhen, Guangdong, Chongqing and Hubei were fully launched; the national unified carbon trading market had started. China needs to learn from the advanced experience of institutional design and platform construction in developed countries and regions, actively introduce carbon trading mechanisms, and explore carbon trading systems. However, due to differences in the development level of various regions and differences in industries, markets and varieties, it is necessary to establish a regulatory department to implement special and unified management of carbon exchanges in various regions.

From an international perspective, carbon trading is an important measure to suppress carbon emissions in Europe and the United States. The suppression of carbon emissions in China's carbon trading is not obvious, On the contrary, it has played a certain role in increasing it. Therefore, China should regulate and develop the carbon trading market, so that it can fully release the potential of carbon emission reduction. The European Union Emissions Trading System (EUETS) is the world's first carbon emissions trading system established in 2005 and still dominates the global carbon trading market. During the period from 2005 to 2013, the growth rate of the European Union Allowance (EUA) trading volume was relatively fast. In 2013, it was as high as 52.348 billion US dollars, and its trading volume and transaction volume are far ahead of the rest of the world, which has greatly improved the low carbon development in European countries. Although China has seven major carbon emissions

exchange, the development level of carbon trading around the country is still low. China should strengthen the norms and focus on the development of carbon trading market, so that it can really play the role of carbon emission reduction.

## 5. Conclusions

In this study, the effect of carbon finance on carbon emissions in the Beijing-Tianjin-Hebei region is examined by using the LMDI method. The empirical results of this study found that the total green credit effect will inhibit the increase of carbon emissions, but the total effect of carbon trading will promote the increase of carbon emissions.

From the above analysis, this study has several policy implications. First, China needs to concentrate resources and strength to create a world-class national carbon trading platform with open and transparent information system and active trading. It is recommended that a special carbon finance regulatory department be established within China's banks to supervise carbon trading in various regions. Second, policy makers are urged to make greater use of the stock market to divert additional funding for clean energy projects and renewable energy projects. Many studies around the world have highlighted the important role of the stock market in reducing carbon emissions [34–38]. Effective and sustainable policies are needed to ensure that all of the listed firms adopt greener technologies in their production activities. Third, on pollution control and energy conservation, on the one hand, the government should take strict measures against highly polluting enterprises, and impose pollution surcharges or carbon taxes onto them. This will encourage them to invest more in clean and renewable energy, which will help greatly reduce carbon dioxide emissions. On the other hand, the government should continue to deepen the green credit field and improve the Green Credit System, further advance the credit management method, implement fiscal and tax incentives for energy conservation and emission reduction and vigorously explore the securitization of green credit assets and absorb more idle assets to support energy conservation and environmental protection. In addition, innovative energy-saving emission reduction technologies and knowledge advancement are inseparable from substantial financial support. It is possible to directly invest in green enterprises through financial funds, and promote the development and application of high technology. It is also possible to increase the scale of investment in environmental protection credit through capital guarantee leverage, and guide and incite social capital to flow into the environmental protection industry. Finally, the constraints of the environmental impact assessment (EIA) should be highlighted. The new high-energy, high emission projects should be strictly constrained by formulated standards, and its sewage intensity and energy efficiency level must reach the domestic advanced level. The main pollutant discharge indicators are regarded as important considerations in the EIA approval, and new capacity for high energy consumption industries such as steel and chemicals. Energy consumption reduction or equivalent replacement should be implemented. EIA approval for new high-energy-consumption projects in areas where energy conservation and emission reduction targets are not completed should be suspended.

**Author Contributions:** Conceptualization, L.L.; Data curation, L.L. and J.H.; Formal analysis, L.L. and D.L.; Methodology, L.L., D.L. and J.H.; Project administration, L.L.; Writing-original draft, L.L. and J.H.; Writing-review & editing, L.L., J.H., W.C. and D.X.

**Acknowledgments:** High tribute is to be paid to the editors and reviewers concerned. This work is supported by grants from National Social Science Foundation of China (15BJL059), National Social Science Foundation of China (17ZDA056), the Ministry of Education Foundation (18JHQ009), and High-level Innovation Team Project Foundation in 2017(IDHT20170505).

**Conflicts of Interest:** The authors declare no conflict of interest. The founding sponsors had no role in the design of the study; in the collection, analyses, or interpretation of data; in the writing of the manuscript, and in the decision to publish the results.

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
