# Peer review of "The Study of the Impact of Carbon Finance Effect on Carbon Emissions in Beijing-Tianjin-Hebei Region—Based on Logarithmic Mean Divisia Index Decomposition Analysis"

_sustainability, doi:10.3390/su11051465_

Round 1

Reviewer 1 Report

Highlight changes in yellow in a next revision, please. No track changes.

Consider comments in the entire text.

Please revise the title using assertive and clear English language: avoid using abbreviations

Abstract: all abbreviations must be defined at first use, here and in the text. Check all

 “LMDI”

Please consider addressing the structure indicated by the journal in instructions

https://www.mdpi.com/journal/sustainability/instructions

Abstract: The abstract should be a total of about 200 words maximum. The      abstract should be a single paragraph and should follow the style of      structured abstracts, but without headings: 1) Background: Place the      question addressed in a broad context and highlight the purpose of the      study; 2) Methods: Describe briefly the main methods or treatments      applied. Include any relevant preregistration numbers, and species and      strains of any animals used. 3) Results: Summarize the article's main      findings; and 4) Conclusion: Indicate the main conclusions or      interpretations. The abstract should be an objective representation of the      article: it must not contain results which are not presented and      substantiated in the main text and should not exaggerate the main      conclusions.

Add brief contextualization

Add quantitative data to reflect results

Avoid referring to the paper as if it was an author

“This paper…

 Decomposes…

 Divides…”

[and later again… “This paper constructs the LMDI”]

Introduction: ONE reference… Not possible and a huge text…

Then I see reference [3] in “2. Literature Review” but no [2], please check

I see no need for a separated literature review… It is not that big and references are still scarce…

Avoid a succession of direct references:

Guo and Pan[17] pointed out

Gu and He[18] believed

Better connect the text.

Avoid expression “many scholars

Looking only at the heading this should then be addressed in literature review...

3. Discussion on Applicability of LMDI Method

It seems in fact to belong to review… “In 1998, Ang[22]first proposed” with ONE reference…

Not enough considering the title…

This should be next to similar information, at the end of the ONE (sections 1 to 3 as only one…) Introduction section

Section 4:

I suggest revising the headings…

4. Constructing an LMDI Model for Carbon Emission Incremental Structure Decomposition

For clarity and assertiveness, as in the title

Address:

italics (parameters) in the text, add units where available

equations format… They differ from case to case…

bold (vectors?),

italics (also check tables…),

font type/size similar to the text, etc

parameters must be defined after EACH equation, not in bulk

All equations must have references immediately before presentation, otherwise they will be considered absolutely original (and then emphasize and highlight novelty), but they are usually based in known equations…

Data sources:” must correspond to reference numbers and included in the final list…

Avoid starting a new paragraph if the text continues, check all:

Specifically, from 2010 to 2016

There cannot be word duplication in different parts of the text, write differently then:

From 2010 to 2016, green credit

From 2010 to 2016, carbon trading

Specifically, from 2010 to 2016, the total contribution

Specifically, from 2010 to 2016, the carbon emission reduction

A scientific text cannot have that…

I see no interest in having that many headings and a tiny text…

5.2. An Empirical Analysis of the Relationship between Carbon Finance and Carbon Emissions in the Sub-region

5.2.1. Contribution of the Total Green Credit Effect of Each Region to Carbon Emissions

A section cannot start with a figure and no previous contextualization

Figure 1. Contribution

Figure 1 and others: outdated style and check all labels in all… axis, also why upper letter, etc?

I would remove the word suggestions in “6. Conclusions and Suggestions

Follow a structure similar to the one recommended above to abstract:

Brief contextualization

Brief methodology

Findings

Practical implications

Enhance novelty

I would not use headings in the Conclusions section, otherwise consider adding them to a Discussion section

A section called Conclusions must be assertive and concise. The reader will not aim to read a big text and be forced to take his/her own conclusions.

Then, the structure of the text must change, and so may headings in such a part of the text makes the reader to get lost.

The quantitative results mentioned through the text and the data treated loose expression here, and everything must be connected.

This part of the text is acting more like a review section

See that headings are mixed with text:

6.3.3. Highlight the Constraints of the EIA Assessment EIA and EIA should be strictly implemented, 474 the new high-energy, high emission projects should be strictly formulated standards, and its

The content of the manuscript is interesting but it needs translation into a relevant text…

Please check same samples

References:

Scarce

They do not follow the guidelines

They need to be updated

Author Response

Thank you for your comments. I will provide point-by-point reply in the form of word document.

Reviewer 2 Report

Title: The Study of the Impact of Carbon Finance Effect on 2 Carbon Emissions in Beijing-Tianjin-Hebei Region—3 —Based on LMDI Method

This is an interesting study. However, authors need to highlight the policy implications which needs to be further strengthened in the paper. I have found some studies at the global context where authors have highlighted the significance of stock markets in promoting renewable and or clean energy projects. These studies also highlighted that the stock markets play an important role in reducing the carbon emissions. Therefore, I advise the authors to strengthen the policy implications on these grounds and can make use of the following references. This will further improve the quality of the manuscript.

1)      Financing clean energy projects through domestic and foreign capital: The role of political cooperation among the EU, the G20 and OECD countries

2)      The effects of stock market growth and renewable energy use on CO2 emissions: Evidence from G20 countries

3)      Financing renewable energy projects in major emerging market economies: Evidence in the perspective of sustainable economic development

4)      The role of stock markets on environmental degradation: A comparative study of developed and emerging market economies across the globe

5)      Determinants of energy demand in African frontier market economies: An empirical investigation

Author Response

Thank you for your comments. I will provide point-by-point reply in the form of word document

Round 2

Reviewer 1 Report

Highlight changes in yellow in a next revision, please. No track changes.

Consider comments in the entire text.

Title: be consistent, upper letter and no “-“ in LMDI. Check:

https://ses.library.usyd.edu.au/bitstream/2123/5329/4/r-wood-2009-thesis.pdf#page=214

Abstract: much better now, although I believe more quantitative data could be included

References are not indicated in the text using the journals correct format, please check instructions and samples, authors cannot mix styles…: Harvard and numbers…

Also, check spacing: “al.,2013”

All over… please… “States’s(BP,

G20.Paramati et al.(2018) [35]

Section 2: change in heading not really significant: “2. A brief overview of literature

Again, avoid using repeated direct references as in

Paramati et al. (2017) [34] confirm

Or “Paramati et al.(2018) [35] considered

Because the text must be fluid and content connected

“logarithmic-mean-divisia-indexLMDI” defined for the first time only in “3. Methodology and Data

But then only abbreviations must be used

form of logarithmic-mean-divisia-index,

Correct it all, because it goes on and on, check all abbreviations use then…

[See that it is included in the TITLE…, it should also be mentioned in ABSTRACT…]

See that equation 2 is much bigger in font size than equation 1 , be consistent and coherent…

Assure that that decomposition is absolutely original then:

Response 8: We have included references of the LMDI method, which is based on the basic addition and multiplication forms, and we have used this basic principle to decompose the green credit effect and the carbon trading effect.

Section 4: revise the diagonal line in the table, above the text…

I would remove the term “relationship” from headings

4.1.1. The Relationship between Green Credit and Carbon Emissions

And more…

Figure 1 and more…

All axis must have a label: check xx axis…

I would remove “policy implications

From “5. Conclusions and policy implications”        

It is included but it does not have to be in the heading

Please remove all “we” and personal expression and avoid the famous “lists”

Instead, better connect all content with relevant language:

Based on the above findings, we draw the following policy implications:

I must say that the conclusions section are not intended to be so extensive, then the reader is forced to take his/her own conclusions, consider moving part of it to results section…

Already mentioned in the previous revision…

References: use DOI as asked by the journal.

Correct formatting and spacing

The text is much better now but it can still be made more relevant.

Consider adding more quantitative information in form of tables or figures and less text.

Author Response

We have uploaded the word document for the specific reply.

Round 3

Reviewer 1 Report

Highlight changes in yellow in a next revision, please. No track changes.

Consider comments in the entire text.

See that the text needs formal corrections…

If abbreviations are defined later in the text, they must be present, and defined…, in abstract too…

logarithmic mean divisia index model

Revise upper letter in “PPM”: parts per million…

Check all first-use abbreviations, as “EPPA”, defined where?

This is not correct English…

Paramati et al. [38] based on their previous research, found

Consider a final revision of the entire language in the text…

Check the “,”?” in “Δ?Δ??Δ??Δ??Δ????

Check the italics in parameters, table 1

Two are still there…

Point 8: Please remove all “we” and personal expression”

Author Response

Please refer to the word document for the specific reply
